# Fractional domain wall statistics in spin chains with anomalous symmetries

**Jose Garre Rubio[1] and Norbert Schuch[1,2]**

**1** University of Vienna, Faculty of Mathematics, Oskar-Morgenstern-Platz 1,
1090 Wien, Austria
**2** University of Vienna, Faculty of Physics, Boltzmanngasse 5,
1090 Wien, Austria

## Abstract

We study the statistics of domain wall excitations in quantum spin chains. We focus on systems with finite symmetry groups represented by matrix product unitaries (MPUs), i.e. finite depth quantum circuits. Such symmetries can be anomalous, in which case gapped phases which they support must break the symmetry. The lowest lying excitations of those systems are thus domain wall excitations. We investigate the behavior of these domain walls under exchange, and find that they can exhibit non-trivial exchange statistics. This statistics is completely determined by the anomaly of the symmetry, and we provide a direct relation between the known classification of MPU symmetry actions on ground states and the domain wall statistics. Already for the simplest case of a $\mathbb{Z}_2$ symmetry, we obtain that the presence of an anomalous MPU symmetry gives rise to domain wall excitations which behave neither as bosons nor as fermions, but rather exhibit fractional statistics. Finally, we show that the exchange statistics of domain walls is a physically accessible quantity, by devising explicit measurement operators through which it can be determined.

 Check for updates

# 1   Introduction

Topological order is a new type of order which defies a description in terms of symmetry breaking and local order parameters; instead, the order in such systems is characterized by a global ordering in their entanglement pattern [1]. In gapped phases, such entanglement order can manifest itself in a variety of phenomena. In two dimensions (2D), topological phases possess ground states which support excitations which can only be created in pairs and are thus topologically protected, and which display non-trivial, including non-Abelian, braiding statistics [2,3]; this behavior arises from the global entanglement in the ground state substrate on top of which those excitations are created.

In one dimension (1D), no topological order exists; in fact, in the absence of symmetries, there is only a single gapped phase. On the one hand, imposing symmetries gives rise to symmetry broken phases. While these differ from topological phases in many ways, they still exhibit topologically protected excitations, namely domain walls between different symmetry broken sectors, which can only be created and destroyed in pairs. Yet, unlike in topologically ordered systems, these excitations do not exhibit any non-trivial statistics, and are simply permuted by the symmetry action. On the other hand, 1D systems with symmetries can also exhibit so-called "symmetry protected topological" (SPT) phases, which possess a unique ground state yet are distinct from the trivial phase: What tells these phases apart is that the physical symmetry acts non-trivially (namely, projectively) on the entanglement [4]. This can be formalized using the language of Matrix Product States (MPS), which form the right framework for the description of ground states of gapped systems [5,6], and which has led to a full classification of SPT phases in 1D [7,8]. In those phases, the projective symmetry action on the entanglement also manifests itself in non-trivial physical properties, such as boundary excitations with fractional charges, or a characteristic counting of multiplicities in the entanglement spectrum.

The landscape of 1D phases gets much more rich if we allow for the inclusion of non-trivial "anomalous" symmetries [9,10]. Anomalous symmetries are symmetries which cannot be expressed as a tensor product of local symmetry actions, and thus cannot be defined locally. Rather, they have to be expressed as a finite-depth quantum circuit, or, equivalently, as the unitary equivalent of an MPS, namely, a Matrix Product Unitary (MPU) [11]. Anomalous symmetries appear naturally at the boundary of 2D SPT phases [12], but they can also show up as symmetries of families of 1D Hamiltonians—we will see such an example momentarily.

The presence of an anomalous symmetry has strong consequences. Most importantly, gapped Hamiltonians with anomalous symmetries cannot have a unique ground state [12,13]:

Rather, they must exhibit symmetry breaking, with the different ground states connected by the symmetry action. Unlike conventional symmetry breaking (i.e., of on-site global symmetries), however, the action of the anomalous symmetry does not simply permute the symmetry broken ground states, but also acts non-trivially on the entanglement in those ground states, which can be classified by the way in which the MPU representation of the symmetry acts irreducibly on the MPS representations of the states [14], generalizing the SPT classification of phases.

This raises the question whether this non-trivial symmetry action has physical, and thus measurable, consequences. In particular, given that those systems have degenerate ground states, and the symmetry acts non-trivially when interchanging them, could this imply that the domain wall excitations carry some signature of this unconventional order? A first step in this direction was taken in Ref. [15], where it was argued that for a specific model with an anomalous $\mathbb{Z}_2$ symmetry, the domain wall excitation in the gapped phase exhibit *semionic* statistics: Under full exchange, they collect a $-1$ phase, and thus, their self-statistics phase is half that of a fermion.

In this paper, we provide a comprehensive and rigorous study the physics of domain wall excitations in gapped phases with anomalous symmetries, and in particular of their exchange statistics. We prove that the statistics of those excitations is completely determined by the anomaly of the symmetry and the way in which it acts on the entanglement in the ground states, as classified in Ref. [14] (and in Ref. [16] outside tensor network methods). Note that these domain wall excitations have a non-trivial statistics regarding the exchange of their creation operators, but that there is no natural notion of braiding in 1D, so this is an aspect in which they differ from anyons in 2D.

In particular, already in the simplest example of an anomalous $\mathbb{Z}_2$ symmetry, this gives rise to domain wall excitations which exhibit a semionic statistics. This is thus showing us a way how anomalous symmetries and their unconventional phases manifest themselves in physically testable properties, rather than having to rely on algebraic invariants of the phases. For completeness, we provide a recipe for how to measure this exchange statistics. Our results are akin to the characterization of symmetry protected topological phases in terms of degenerate edge modes and the degeneracies of their the entanglement spectrum, rather than through projective representations of the symmetry group. Altogether, our work provides an extension of the classification of quantum phases under anomalous symmetries beyond ground state properties, by connecting them to the behavior of low-lying excited states.

We start our paper by discussing in detail the simplest case of an anomalous $\mathbb{Z}_2$ symmetry where we deepen on the semionic statistics of the domain walls found in the deformed Ising Hamiltonian studied in Ref. [15] and we compare with the on-site case. To be more tangible, these two cases are manifested in the family of Hamiltonians $H_p(\mu)$ where $p = 0, 1$:

$$H_p(\mu) = \sum_i -X_i(-CZ_{i-1,i+1})^p - \mu Z_i Z_{i+1}, \tag{1}$$

where the symmetry is $U_p = \prod_i X_i \prod_i (CZ_{i,i+1})^p$. Here, $CZ_{i,j}$ is the controlled-Z gate between sites $i$ and $j$ (which applies a $-1$ phase exactly on $|k\rangle_i |l\rangle_j$ if $k = l = 1$). The Hamiltonian $H_0(\mu)$ is the transverse-field Ising model that hosts an ordered phase (ferromagnetic ordering) for $\mu > 1$. The Hamiltonian $H_1(\mu)$ has a two-fold degenerate ground state with a gap above for $\mu > 0$ [15].

We then generalize our findings to gapped systems with a finite symmetry group represented by MPUs, by modeling their ground space by MPS. In that case, we analyze both the domain wall statistics in case where the symmetry is fully broken (see also [17]), as well as other situations where the symmetry is only partially broken, where we find projective actions of the symmetry on the domain walls, see Refs. [18–21]. Furthermore, our results allow us to connect these projective actions with the invariants classifying MPU-symmetric phases [14,16].

Finally, we show that the non-trivial domain wall statistics which we derived are physically accessible quantities: We provide an explicit operator which one can measure to detect the non-trivial exchange statistics of the domain wall excitations.

## 2 Tensor network basics

### 2.1 Matrix product states and the fundamental theorem of MPS

A Matrix Product State (MPS) on a spin chain of length $n$ with local Hilbert space dimension $d$, $(\mathbb{C}^d)^{\otimes N}$, is defined via a $d \times D \times D$ tensor $A$, where $D$ is called the *bond dimension* or *virtual dimension*; the corresponding space is called *virtual space*. We consider MPSs that are translationally invariant where each tensor in the MPS is the same. A $N$-site periodic boundary condition MPS with tensor $A$ is defined by

$$|\psi_A\rangle = \sum_{\{i_k\}} \text{Tr}[A^{i_1}\cdots A^{i_N}]|i_1,\ldots,i_N\rangle,$$

where $A = \sum_i |i\rangle \otimes A^i$ and $A^i$ is a $D \times D$ matrix for each $i = 1,\ldots,d$.

An important class of MPSs, dubbed injective MPS, corresponds to the ones whose tensors $A$ – interpreted as a map from the virtual to the physical system – posses a left-inverse, that is, $\hat{A}A = \mathbb{1}_{D\times D}$. To any injective MPS a gapped local and frustration free Hamiltonian, called *parent* Hamiltonian, can be associated whose unique ground state is the injective MPS.

One can also consider block-injective tensors $A^i = \bigoplus_{x\in\mathcal{I}} A^i_x$, where each $A^i_x$ is a $D_x \times D_x$ matrix and $x \in \mathcal{I}$ denotes the block labels – that is, for each $x$, the $A^i_x$ define an injective MPS. Then, the ground space of the corresponding parent Hamiltonian is spanned by the individual blocks of the MPS, $\{|\psi_{A_x}\rangle, x \in \mathcal{I}\}$, so that its degeneracy coincides with the number of blocks. Note that any MPS can be brought into a block-injective form [22] after blocking a *small* number of sites [23]. Because any two injective MPSs become either orthogonal or proportional in the thermodynamic limit [24], when considering degenerate ground spaces spanned by injective MPSs we restrict w.l.o.g. to blocks which become orthogonal.

For example, the MPS tensor $A^0 = |0\rangle\langle0|, A^1 = |1\rangle\langle1|$ corresponds to the GHZ state and its parent Hamiltonian has a two-fold degenerate ground space, spanned by $|00\cdots0\rangle$ and $|11\cdots1\rangle$. This degeneracy can be seen as symmetry breaking: there is always a symmetry action that permutes between the ground states, i.e. between the different blocks of the MPS. In the previous example, the symmetry is just $\sigma_x^{\otimes n}$.

An important question regarding MPSs is the remaining gauge freedom of the tensors when constrcuting a specific state – that is, if for two MPS tensors $A$ and $B$, $|\psi_A\rangle = |\psi_B\rangle$, then how $A$ and $B$ are related? The so-called 'Fundamental Theorem (FT) of MPSs' answers this question – in fact, depending on the assumptions on the given tensors, different results have been stated [25–27]. In this work, we will rely on the FT proven in Ref. [26], which informally[1] states the following:

**Theorem 1** (Fundamental Theorem of MPS of Ref. [26])**.** *If A is injective and* $|\psi_A\rangle = |\psi_B\rangle$*, and where no assumption on B is required, there exists a pair of matrices* $(V, \hat{V})$ *acting on the virtual level such that*

$$VB^i\hat{V} = A^i, \quad B^i\hat{V}A^jVB^k = B^iB^jB^k, \quad V\hat{V} = \mathbb{1}_{D_A}.$$

*Moreover if there are two such pairs* $(V, \hat{V})$ *and* $(W, \hat{W})$ *then they are related by a constant factor:*

$$VB^i = \lambda \cdot WB^i, \quad B^i\hat{V} = \lambda^{-1} \cdot B^i\hat{W} \quad \Rightarrow \quad VB^i\hat{W} = \lambda \cdot A^i.$$

---

[1]We state the theorem assuming that the nilpotency length of the off-diagonal blocks of $B$ is 1, see [26] for details; this can always be achieved by blocking.

## 2.2 Matrix product unitaries

The global symmetries we consider in this work have the structure of a matrix product unitary (MPU). MPUs are unitary operators, analogous to MPSs, that can be constructed using local tensors for any system size. We define a periodic boundary condition MPU on a chain $(\mathbb{C}^d)^{\otimes N}$ with bond dimension $\chi$ by placing a $d \times d \times \chi \times \chi$ tensor $T$ at every site:

$$U = \sum_{\{i_k, j_k\}} \mathrm{Tr}[T^{i_1 j_1} T^{i_2 j_2} \cdots T^{i_N j_N}] |i_1 \cdots i_N\rangle\langle j_1 \cdots j_N|,$$

where $i, j = 1, \ldots, d$ and $T^{i,j}$ is defined through $T = \sum_{i,j} |i\rangle\langle j| \otimes T^{i,j}$. Interestingly, for MPUs, the unitarity condition implies that the tensors can be chosen to be injective [11]. We note that regular tensor product of unitaries, $U = u^{\otimes N}$, are indeed MPUs with trivial bond dimension $\chi = 1$. We will consider global symmetries coming from a finite group and represented by MPUs: $U_g U_h = U_{gh}$, $g, h \in G$.

We emphasize that Theorem 1 can in particular be applied to the product of two MPUs, as well as to the MPU action on an MPS; this is done simply by 'vectorizing' the MPUs to MPSs. As we will see, this will be a powerful tool which will allow us to derive local characterizations of global properties.

As customary when working with tensor networks we will use a graphical notation to denote the tensors such that MPSs and MPUs are depicted as follows (for details, see e.g. [13]):

$$|\psi_A\rangle = \underset{A \ A}{\boxed{\quad}} \cdots \underset{A}{\rightarrow}, \quad U = \underset{}{\boxed{\overset{T}{\bullet} \overset{T}{\bullet} \cdots \overset{T}{\bullet}}}.$$

# 3 $\mathbb{Z}_2$ MPU representation and its domain walls

## 3.1 Anomaly index of the MPU symmetry and fusion tensors

We consider an MPU

$$U = \sum_{\{i_k, j_k\}} \mathrm{Tr}[T^{i_1 j_1} T^{i_2 j_2} \cdots T^{i_N j_N}] |i_1 \cdots i_N\rangle\langle j_1 \cdots j_N| = \underset{}{\boxed{\overset{T}{\bullet} \overset{T}{\bullet} \cdots \overset{T}{\bullet}}}$$

of order two, $U^2 = \mathbb{1}$, for any length $n$. Since the tensor $T$ can be chosen to be injective [11], the global condition $U^2 = \mathbb{1}$ implies that a *local* condition must hold (which in turn gives rise to the global one) using Theorem 1: There exists a pair of *fusion tensors* $(V, \widehat{V})$ acting at the virtual level which satisfy [26]

$$\overbrace{\phantom{xxxxx}}^{m+1} = \overbrace{\phantom{xxx}}^{m+1} \quad \text{and} \quad \overbrace{\phantom{xxx}}^{m} = \overbrace{\phantom{xx}}^{m} \qquad (2)$$

for all $m \geq 0$. Notice that the fusion tensors are defined up to a phase factor $V \to \beta V$ and $\widehat{V} \to \beta^{-1}\widehat{V}$.

Since there are two local ways to decompose the product of 3 tensors $(TT)T = T(TT)$ using the fusion tensors, these two decomposition are related by a phase factor:

$$\boxed{\phantom{xxx}} = \omega \cdot \phantom{x}\bullet\bullet\bullet\phantom{x} \qquad (3)$$

where $\omega$ is invariant under phase factor redefinitions of the fusion tensors, i.e. it is a gauge invariant quantity. By applying the above equation in two different ways to the product of 4 tensors:

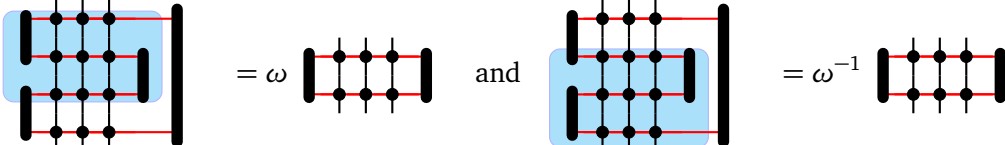

we obtain that $\omega$ has to satisfy the consistency equation

$$\omega^2 = 1, \tag{4}$$

and thus, $\omega = \pm 1$.

Notice that for an on-site symmetry of order two, $U = u^{\otimes N}$ with $u^2 = \mathbb{1}$, we have that $T^{ij} = u_{ij}$, i.e., the virtual dimension is 1, and thus the fusion tensors defined in Eq. (2) are trivial. Then, on-site symmetries satisfies $\omega = 1$. In this work we are interested in the case where $\omega = -1$ – we refer to this case as *anomalous*, and say that the MPU carries an anomaly.

## 3.2 Ground space and domain walls

Let us now turn towards the structure of ground states of local, gapped Hamiltonians $H$ which possess an MPU symmetry, $HU = UH$. As is customary, we will assume that the ground space of the Hamiltonian is spanned by injective Matrix Product States (MPS): This is well motivated by the fact that MPS faithfully approximate ground states of gapped local Hamiltonians [5,6], and once one has a MPS basis of the ground space, one can decompose the MPS into injective blocks [22], each of which is again a ground state. Since the Hamiltonian is symmetric under the MPU $U$, the ground space must be left invariant by the action of $U$ as well. These properties – symmetric ground space spanned by injective MPS – is all which we will make use of in what follows, and no other information on the Hamiltonian will be required.

In the case of an anomalous MPU, an important point arises: As was shown in Ref. [12], an anomalous MPU cannot leave an injective MPS invariant. This implies that the ground space must be at least two-fold degenerate (which it will be in the absence of additional symmetries or fine-tuning), spanned by two MPS $|\psi_A\rangle$ and $|\psi_B\rangle$ with MPS tensors $A$ and $B$, respectively, on which $U$ acts by interchanging them,[2]

$$U|\psi_A\rangle = |\psi_B\rangle. \tag{5}$$

Just as before for MPUs, the global symmetry in Eq. (5) implies (using Theorem 1) that a *local* equation which characterizes this symmetry action holds: There exist a pair of *action tensors* $(W_B, \widehat{W}_B)$ acting at the virtual level that satisfy for all $m \geq 0$

$$\tag{6}$$

---

[2]More generally, we could allow for $U|\psi_A\rangle = c_N|\psi_B\rangle$ for length $N$, $|c_N|^2 = 1$. Since $|\psi_B\rangle$ is injective, the canonical form of $U|\psi_A\rangle$ consists of a fixed number of blocks $\tilde{A}_i$ which are all proportional to $B$, $\tilde{A}_i = \lambda_i X_i B X_i^{-1}$, and thus $c_N = \sum_i \lambda_i^N$. By choosing an $N$ such that all $\lambda_i^N$ are sufficiently close to the positive real axis, and using $|\sum \lambda_i^m| = 1$ for $m = N, 2N$, we find that there can only be a single $\lambda_i \neq 0$, which must be a phase. Thus, $U|\psi_A\rangle = \lambda^N|\psi_B\rangle$ with some phase $\lambda$ which can be absorbed in the tensor $B$, leaving us with Eq. (5).

where we have the freedom to redefine the action tensors by a phase factor $W_B \to \gamma_B W_B$ and $\widehat{W}_B \to \gamma_B^{-1} \widehat{W}_B$. Analogously, the equation $U|\psi_B\rangle = |\psi_A\rangle$ gives rise to the pair of action tensors $(W_A, \widehat{W}_A)$ which satisfy an equation analogous to Eq. (6).

In a system with more than one ground state, the natural fundamental excitations are domain walls between the different ground states. To model those domain walls for the setting at hand, we introduce "domain wall tensors" $e_{AB}$ and $e_{BA}$ which interface between domains with $A$ and $B$ tensors as

$$-\!-\underset{A\ \ A\ e_{AB}\ B\ \ B}{\text{⦁⦁■⦁⦁}}-\!- \quad \text{and} \quad -\!-\underset{B\ \ B\ e_{BA}\ A\ \ A}{\text{⦁⦁■⦁⦁}}-\!- \ . \tag{7}$$

Intuitively, the $e_{AB}$ and $e_{BA}$ are meant to model the fundamental excitations of the Hamiltonian, but any choice will do, as long as they are interchanged by the symmetry action. Let us define what we mean by that more specifically: Given domain wall tensors $e_{AB}, e_{BA}$, we can construct a state on a periodic system with a pair of domain walls at some specific positions $k$ and $\ell$,

$$|\psi(A-B-A)\rangle = \sum_{\{i_k\}} \text{Tr}[A^{i_1} \cdots A^{i_{k-1}} e_{AB}^{i_k} B^{i_{k+1}} \cdots B^{i_{\ell-1}} e_{BA}^{i_\ell} A^{i_{\ell+1}} \cdots A^{i_N}]|i_1 \cdots i_k \cdots i_\ell \cdots i_N\rangle$$

$$= \underset{A\ \ A\ e_{AB}\ B\ \ B\ e_{BA}\ A\ \ A}{\boxed{\text{⦁⦁■⦁⦁■⦁⦁}}} \tag{8}$$

as well as its "twin"

$$|\psi(B-A-B)\rangle = \underset{B\ \ B\ e_{BA}\ A\ \ A\ e_{AB}\ B\ \ B}{\boxed{\text{⦁⦁■⦁⦁■⦁⦁}}} \ . \tag{9}$$

We now demand that the symmetry acts by exchanging these two states (up to a phase), $U|\psi(A-B-A)\rangle = c|\psi(B-A-B)\rangle$. Using the injectivity of $A$ and $B$ and the relations in Eq. (6) we obtain:

$$-\underset{\widehat{W}_B\ A\ e_{AB}\ B\ W_A}{\boxed{\phantom{xx}}}\ \otimes\ -\underset{\widehat{W}_A\ B\ e_{BA}\ A\ W_B}{\boxed{\phantom{xx}}}\ =\ c \cdot \underset{B\ e_{BA}\ A}{\text{⦁■⦁}}\ \otimes\ \underset{A\ e_{AB}\ B}{\text{⦁■⦁}}\ . \tag{10}$$

That in particular entails how the domain walls transform into each other since this implies that

$$-\underset{\widehat{W}_B\ A\ e_{AB}\ B\ W_A}{\boxed{\phantom{xx}}}\ =\ c_{AB}\underset{B\ e_{BA}\ A}{\text{⦁■⦁}} \quad \text{and} \quad -\underset{\widehat{W}_A\ B\ e_{BA}\ A\ W_B}{\boxed{\phantom{xx}}}\ =\ c_{BA}\underset{A\ e_{AB}\ B}{\text{⦁■⦁}} \tag{11}$$

must hold, where $c_{AB}$ is defined up to $\gamma_B/\gamma_A$ due to the dependence of $W_A$ and $W_B$ on their phase factor choice. In fact, in order to ensure that the domain walls transform into each other under the symmetry, we could have chosen any $e_{AB}$ and then *defined* $e_{BA}$ through the left part of Equation (11), for an arbitrary choice of $c_{AB}$. Note, however, that the choice of $c_{AB}$ fixes $e_{AB}$ and $e_{BA}$ (relative to $\gamma_B/\gamma_A$), and in particular their relative phase, and thus $c_{BA}$ is uniquely determined.

In fact, from Eq. (11), one can easily check that the product $c_{BA}c_{AB}$ is gauge invariant. This can also be understood more directly, by observing that $U|\psi(A-B-A)\rangle = c_{BA}c_{AB}|\psi(B-A-B)\rangle$, that is, $c_{BA}c_{AB} = c$ is a physically observable quantity. Since $U^2 = \mathbb{1}$, it must hold that $(c_{BA}c_{AB})^2 = 1$, and therefore, the way the symmetry acts on domain walls must fall into one of the two cases

$$c_{BA}c_{AB} = \pm 1 \, . \tag{12}$$

The fact that both $c_{BA}c_{AB}$ and the phase $\omega$ [Eq. (3)] which characterizes the anomaly of the MPS can take values $\pm 1$ is not a coincidence. As we will show in the next subsection, they are in fact equal, $c_{AB}c_{BA} = \omega$: That is, the anomaly of the MPU symmetry shows up in the transformation properties of domain wall excitations.

Translation symmetry allows to construct excitations with momentum $k$ by a superposition of the domain walls in Eq. (7) as in [28]. We note that the MPU action on those well-defined momenta excitations would result in the same phase factors as in Eq. (11).

### 3.3 Relation between $c_{BA}c_{AB}$ and the MPU anomaly

Let us now consider

$$(U^2)|\psi_A\rangle = U(U|\psi_A\rangle) = |\psi_A\rangle. \tag{13}$$

As $|\psi_A\rangle$ is an injective MPS, the left equality once more implies that a corresponding local condition holds, where for the left term, we first merge the two MPU tensors $T$ using Eq. (2) (the resulting MPU $U^2 = \mathbb{1}$ is trivial, so it merges trivially with $A$), while for the right term, we merge the MPU tensors $T$ one after another with the MPS tensor $A$, using Eq. (6). The two decompositions are then related via a phase factor $L_A$ [which measures the associativity of the fusion orders $(T \cdot T) \cdot A$ and $T \cdot (T \cdot A)$],

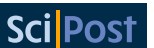

$$= L_A \tag{14}$$

Note that $L_A$ is only defined up to a phase factor $\gamma_A \cdot \gamma_B/\beta$, arising from the gauge freedom in the action and fusion tensors. In the same way, $L_B$ can be defined. We will now derive a key relation between these $L$-symbols and $\omega$ which characterizes the anomaly of the MPU in Eq. (3), namely

$$L_A/L_B = \omega, \tag{15}$$

which is invariant under phase factor redefinitions of the tensors. This relation can be shown as follows (we start by simplifying the blue region):

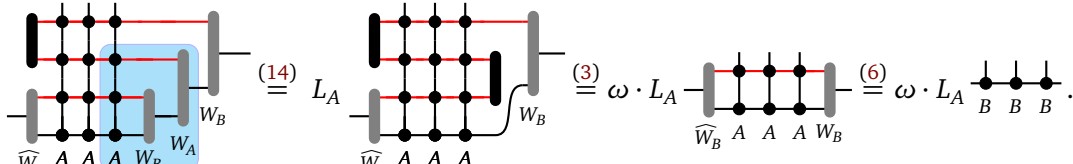

On the other hand, we can transform the left hand side in the above equation by starting to simplify the bottom left part, again marked blue:

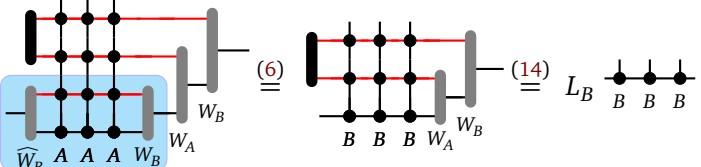

Comparing the two equations, we indeed find $L_A/L_B = \omega$.

Finally, we have that

$$
c_{BA}c_{AB} \quad = \quad = \frac{1}{L_A} \quad = \frac{L_B}{L_A} \quad = \frac{L_B}{L_A} \tag{16}
$$

that is,

$$c_{BA} \cdot c_{AB} = L_A/L_B = \omega. \tag{17}$$

We thus find the product $c_{AB}c_{BA}$, which characterizes how pairs of domain walls transform under the MPU symmetry, is precisely given by the phase $\omega$ which characterizes the anomaly of the MPU – differently put, the MPU anomaly can be physically probed through the way in which it acts on domain wall excitations.

Let us finally point out another connection established by Eq. (17): It links the $c$'s, which characterize the symmetry action on domain wall excitations, to the $L$'s, which characterize the way in which the symmetry acts on the ground state. Those $L$'s have been introduced in Ref. [14], where it was shown that their equivalence classes are the invariants which classify quantum phases with MPU symmetries. Importantly, for a $\mathbb{Z}_2$ symmetry, the quotient $L_A/L_B$ completely characterized the quantum phase of a given Hamiltonian.

## 3.4   The simplest example

Let us now illustrate this through a simple example. To this end, consider the two states $|0\rangle^{\otimes N}$ and $|1\rangle^{\otimes N}$ on $(\mathbb{C}^2)^{\otimes N}$, and two different $\mathbb{Z}_2$ symmetries which permute them: First, a trivial local symmetry $U_x = \prod_i X_i$, and second, the CZX symmetry $U_{CZX} = \prod_i Z_i \cdot CZ_{i,i+1} \prod_i X_i$, which has an anomaly, i.e., $\omega = -1$ [12] (here, $CZ_{i,i+1}$ is the controlled-Z on qubits $i$ and $i+1$, i.e., it applies a $-1$ phase exactly if both qubits are in the state $|1\rangle$). For the trivial symmetry action, it is easy to see that the fusion tensors for $U_x$ are trivial, so $\omega = +1$ and since $X$ swaps $|0\rangle$ and $|1\rangle$, the action tensors are trivial too, so $L_0/L_1 = +1$.

Let us now consider the CZX MPU $U_{CZX}$. We will now calculate the fusion tensors of $U_{CZX}$, as well as the action tensors when applied to $|0\rangle^{\otimes N}$ and $|1\rangle^{\otimes N} = U_{CZX}|0\rangle^{\otimes N}$, to show that in this case $\omega = L_0/L_1 = -1$. To this end, notice that the $CZ$ gate can be written as a tensor network using delta tensors and (unnormalized) Hadamard matrices as follows:

$$CZ = \left|\begin{array}{c}\hat{H}\\\bullet\end{array}\right|, \quad \hat{H} = \begin{pmatrix} 1 & 1 \\ 1 & -1 \end{pmatrix}, \quad \left|\begin{array}{c}j\\\text{---}k\\i\end{array}\right. = \delta_{ijk}.$$

The MPU tensor of $U_{CZX}$ that we choose is

$$\begin{array}{c}\text{---}\bullet\text{---}\end{array} = \begin{array}{c}Z\\\hat{H}\bullet\\X\end{array}.$$

Notice that this tensor becomes injective when blocking two sites. To obtain the fusion tensor we only need to use 3 sites as follows:

$$\begin{array}{c}\bullet\bullet\bullet\\\bullet\bullet\bullet\end{array} = \begin{array}{c}V\end{array} \quad \Big| \quad \begin{array}{c}\hat{V}\end{array} = (-1)\begin{array}{c}\hat{H}\\\bullet\bullet\\\hat{H}X\end{array} z \Big| z \begin{array}{c}\bullet\\X\end{array}, \quad \begin{array}{c}\end{array} = \begin{array}{c}\langle 1|\\\langle 0|\end{array}, \quad \begin{array}{c}\end{array} = \begin{array}{c}-|\hat{-}\rangle\\|\hat{+}\rangle\end{array}$$

where $|\hat{+}\rangle = |0\rangle + |1\rangle$ and $|\hat{-}\rangle = |0\rangle - |1\rangle$.

Since the MPSs we are dealing with have bond dimension $D = 1$, i.e., they are product states, the fusion tensors have only one index coming from the MPU virtual level – that is, they are vectors. Then, we obtain the following decompositions:

$$\begin{array}{c}\bullet\bullet\bullet\\|0\rangle|0\rangle|0\rangle\end{array} = \begin{array}{c}|\hat{+}\rangle\\\bullet\\|0\rangle\end{array} \begin{array}{c}\langle\hat{+}|\\\bullet\\|1\rangle\end{array} \begin{array}{c}\\\\|0\rangle\end{array}, \quad \begin{array}{c}\bullet\bullet\bullet\\|1\rangle|1\rangle|1\rangle\end{array} = \begin{array}{c}-|\hat{+}\rangle\\\bullet\\|1\rangle\end{array} \begin{array}{c}\langle\hat{+}|\\\bullet\\|0\rangle\end{array} \begin{array}{c}\\\\|1\rangle\end{array}.$$

We can now obtain the $L$-symbols defined in Eq. (14) (and using Eq. (2)):

$$\begin{array}{c}\\\bullet\\|i\rangle\end{array} = L_i \cdot \begin{array}{c}\bullet\\\bullet\\|i\rangle\end{array} = L_i \begin{array}{c}-Z\\\bullet\\|i\rangle\end{array} = (-1)^{i+1}L_i \cdot \begin{array}{c}\\\bullet\\|i\rangle\end{array} \Rightarrow L_i = (-1)^{i+1} \Rightarrow L_0/L_1 = -1.$$

### 3.5  Creation of domain walls

Let us now study how to create a pair of domain wall excitations on top of an (unkown) ground state, located at positions $i$ and $j$. In the case of an on-site symmetry, $U = u^{\otimes N}$, this can be accomplished by acting with the symmetry operation $\bigotimes_{x \in [i,j]} u^{[x]}$ on the region $[i, j]$. In addition, if we are interested in creating a "nice" excitation, such as an eigenstate of the Hamiltonian, we will have to dress the endpoints, that is, act with an additional operator around sites $a$ and $b$. The support of this operator will, in general, depend on the correlation length of the system: While for an RG fixed point, an operator with fixed support will create an exact eigenstate, for a system away from the fixed point the accuracy of the eigenstate will converge exponentially with the ratio of its support $\ell$ and the correlation length $\xi$. Note that in MPS, a similar effect can achieved by changing the tensor at sites $i$ and $j$, since this affects the physical state on a scale of the correlation length.

In the following, we will focus on the case of the RG fixed point for clarity. However, the same analysis applies to MPS with finite correlation $\xi$ length away from the fixed point: By blocking $\ell$ tensors, one obtains an MPS whose tensors converge to an RG fixed point exponentially in $\ell/\xi$.

In the case of an MPU symmetry, we can't simply act with a bare symmetry string: There are open virtual indices at the ends of the MPU string operator, which we need to take care of. We do so by adding three-leg endpoint tensors (yellow diamonds) at the two ends of the MPU string:

$$|\psi(A{-}B{-}A)\rangle = O^{[i,j]}|\psi_A\rangle = \; \cdots \; \begin{array}{c} \end{array} \; \cdots \tag{18}$$

Here, the endpoint tensors are chosen precisely such as to create the previously constructed endpoint excitations of Eqs. (8) and (9) when acting on $|\psi_A\rangle$ and $|\psi_B\rangle$, respectively. This is achieved by defining

$$\tag{19}$$

and correspondingly for the right endpoint, where $\hat{A}$, $\hat{B}$ are the left inverses of the MPS tensors $A$ and $B$, respectively – those exist since $|\psi_A\rangle$ and $|\psi_B\rangle$ are injective, and have orthogonal support since we work at the RG fixed point (otherwise, we would have to define them to act on a block of $\ell$ tensors), and thus the two terms in the sum act separately on $A$ and $B$, respectively.

### 3.6  Exchange statistics of domain walls

Let us now turn to our key goal: To characterize the exchange statistics of domain wall excitations. To this end, we need to consider individual domain walls, which we obtain by cutting the operator creating a pair in the center,

$$O_x^{[i]} = \quad \begin{array}{c} i \\ \diamond{-}\bullet{-}\bullet{-}\bullet \end{array} |x\rangle \; , \tag{20}$$

where we fix the open MPU bond to have a value $x$ (which captures the correlations between left and right domain wall.) Acting with $O_x^{[i]}$ on the left half of $|\psi_A\rangle$ results in

where we have used (19) and (6). The resulting object has a cut on the right, with an $x$-dependent dangling MPU leg, which needs to be re-glued with the other half of the domain wall creation operator to make sense as a state, and – using again Eq. (6) – results in the pair of domain wall excitations Eq. (18). For simplicity, we denote the resulting object (which in principle is defined only on a half-chain) by $O_x^{[i]}|\psi_A\rangle$.

We can now study the exchange statistics of domain walls by comparing $O_x^{[i]}O_y^{[j]}$ and $O_x^{[j]}O_y^{[i]}$. This can be interpreted as exchanging the position of the two domain walls, while leaving the order of their creation unchanged (that is, as exchanging $i$ and $j$). Alternatively, this can be interpreted as the effect of exchanging the order in which the domain walls at positions $i$ and $j$ are created, while keeping the order at the other (right) end unchanged (that is, exchanging the ordering of $O^{[i]}$ and $O^{[j]}$, together with exchanging their $x$ and $y$). In either case, the MPU labels $x$ and $y$ at the right cut are left untouched, which allows to study the effect at the two ends independently. For $i > j$, this results in

$$O_x^{[j]}O_y^{[i]}|\psi_A\rangle = \quad = $$

$$= c_{AB} \qquad = c_{AB}$$

$$= c_{AB}\, O_x^{[i]}O_y^{[j]}|\psi_A\rangle\,,$$

i.e., we obtain the commutation relation

$$O_x^{[j]} \cdot O_y^{[i]}|\psi_A\rangle = c_{AB}\, O_x^{[i]} \cdot O_y^{[j]}|\psi_A\rangle\,, \quad i > j\,. \tag{21}$$

That is, $c_{AB}$ is the phase factor resulting from the interchange of two domain walls on top of the ground state $|\psi_A\rangle$. Analogously, $c_{BA}$ results from the interchange of domain walls on top of $|\psi_B\rangle$.

What is the statistics of these domain wall excitations? Let us first repeat the above calculation with three domain walls at positions $i > j > k$, which we exchange twice:

$$O_z^{[k]}O_y^{[j]}O_x^{[i]}|\psi_A\rangle = c_{AB} \cdot O_z^{[k]}O_y^{[i]}O_x^{[j]}|\psi_A\rangle = \overbrace{c_{AB}c_{BA}}^{\omega} \cdot O_z^{[i]}O_y^{[k]}O_x^{[j]}|\psi_A\rangle\,, \quad i > j > k\,, \tag{22}$$

which can be seen as the composition of the transposition operators $T_{12} \circ T_{23}$. Such a double exchange would give a phase of $+1$ for both bosons and fermions. In the case of an anomalous MPU symmetry with $\omega = -1$, such as $U_{CZX}$, this will however result in a phase $-1$ – that is, the domain walls behave neither as bosons nor as fermions. Rather, they exhibit a fractional *semionic* statistics, where exchanging twice gives rise to a minus sign. In fact, since we are free to fix $c_{AB}$ by suitably defining $e_{BA}$ [see discussion below Eq. (11)], we can choose $c_{AB} = c_{BA} = i$, in which case the interchange of two domain walls will result in a phase $i$, i.e., semionic statistics. We note the same conclusion has been found in Ref. [29] by also truncating the symmetry operators.

We emphasize that the gauge invariant quantity $c_{AB}c_{BA} = \omega$ can be physically measured by using the operators defined in (18). To do so, we consider two pairs of domain wall placed on sites $(i_1, j_1)$ and $(i_2, j_2)$ satisfying $i_2 < i_1 < j_1 < j_2$, for which

$$O^{[i_1, j_1]}O^{[i_2, j_2]}|\psi_A\rangle = c_{AB}c_{BA} \cdot O^{[i_2, j_2]}O^{[i_1, j_1]}|\psi_A\rangle\,. \tag{23}$$

# 4 General case

In this section, we generalize the study done in the previous section to the case of a global symmetry given by an MPU representation of a finite group $G$. To do so we first explain what is the generalization of $\omega$, the anomaly of an MPU representation of $G$. Second, we revisit how the action of those MPUs act on ground states and we also explain how the classifying invariants of these phases are constructed (generalizing $L_A$ and $L_B$). Then, we define quantities characterizing the action of the MPUs on the domain wall excitations (generalizing $c_{AB}$ and $c_{BA}$) and we connect these to the invariant of the quantum phases. Finally we show how these quantities are related to the interchange of domain wall excitations.

## 4.1 MPU symmetries and their invariant ground states

We consider MPU representations of a finite group $G$: $U_g U_h = U_{gh}$ and $U_e = \mathbb{1}$ given by injective tensors. Using Theorem 1, this implies that for every pair $g, h \in G$ of group elements, there are pairs of *fusion* tensors $(V_{g,h}, \widehat{V}_{g,h})$ acting at the virtual level that satisfy:

$$\tag{24}$$

for all $m \geq 0$, where we omit the label $V_{g,h}$ of the fusion tensors and we just indicate the group element where they act. Notice that Eq.(24) is invariant under the redefinition of the fusion tensors by phase factors $V_{g,h} \to \beta_{g,h} V_{g,h}$ and $\widehat{V}_{g,h} \to \beta_{g,h}^{-1} \widehat{V}_{g,h}$.

Using Theorem 1, different fusion tensor decompositions are related by a phase factor $\omega(g, h, k)$:

$$\tag{24}$$

and it can be shown by decomposing in different ways the product of 4 MPU tensors, that $\omega$ satisfies a consistency equation [26]: The so-called 3-cocycle condition:

$$\omega(g,h,k)\omega(g,hk,l)\omega(h,k,l) = \omega(gh,k,l)\omega(g,h,kl). \tag{25}$$

Due to the phase factor freedom on the definition of the fusion tensors, the 3-cocycles are defined up to a phase $\beta_{h,k}\beta_{g,hk}/\beta_{g,h}\beta_{gh,k}$ which correspond to a 3-coboundary. Quotienting by this freedom, 3-cocycles are classified by the third cohomology group of $G$, $\mathcal{H}^3[G, U(1)]$, a finite abelian group. For example, the case studied in the previous section results in $\mathcal{H}^3[\mathbb{Z}_2, U(1)] = \mathbb{Z}_2$ which means that there are only two classes of 3-cocycles; the trivial one and the anomalous one.

We represent the ground space by a set of injective MPS $\{|\psi_x\rangle, x \in \mathcal{I}\}$, permuted transitively by the MPU representation of $G$ such that for every $x, y \in \mathcal{I}$ there is a $g \in G$ satisfying $gx = y$ and $U_g|\psi_x\rangle = |\psi_y\rangle$. That implies [26] the existence of a set of action tensors which locally reduce the action of the MPU tensors on the MPS tensors in the following way:

$$\tag{26}$$

for all system sizes $m \geq 0$. Eq.(26) is invariant under the redefintion of the action tensors by a phase factor $W_{g,x} \to \gamma_{g,x} W_{g,x}$ and $\widehat{W}_{g,x} \to \gamma_{g,x}^{-1} \widehat{W}_{g,x}$. The local action of the MPU on the MPS

is associative up to a phase factor, the so-called $L$-symbols:

$$g \quad \begin{matrix} g \\ h \\ gh \end{matrix} \quad y = L_{g,h}^x \quad \begin{matrix} g \\ h \\ x \end{matrix} \quad y$$

which satisfy the following equation involving the 3-cocycle of the MPU symmetry:

$$L_{h,k}^x \cdot L_{g,hk}^x = \omega(g,h,k) \cdot L_{g,h}^{k \cdot x} \cdot L_{gh,k}^x. \tag{27}$$

The solutions to the previous equation, up to the equivalence relation $L_{g,h}^x \sim L_{g,h}^x \frac{\gamma_{h,x}\gamma_{g,hx}}{\gamma_{gh,x}\beta_{g,h}}$, classify the different phases protected by MPU symmetry representation of $G$ given by $\omega$ [14]. Let us review this classification. A MPU with a non-trivial 3-cocycle restricts the possible degeneracies of the ground state of a symmetric Hamiltonian. To see this let us denote by $H \subseteq G$ the unbroken symmetry group ($h \cdot x = x$ for all $h \in H$ and $x \in \mathcal{I}$) such that the ground state degeneracy is $|G/H|$. Then, Eq.(27) restricted to $H$ is $L_{h_2,h_3}^x \cdot L_{h_1,h_2h_3}^x = \omega(h_1,h_2,h_3) \cdot L_{h_1,h_2}^x \cdot L_{h_1h_2,h_3}^x$ which implies that $\omega$ is trivial in $H$ (there is a gauge freedom such that $\omega|_H = 1$). Given $\omega$, we denote by $\hat{H}_\omega \subseteq G$ the biggest subgroup where this can happen, so the smallest ground state degeneracy is $|G/\hat{H}_\omega|$.

Then, the possible phases of Hamiltonians symmetric under MPUs indexed by $(G, \omega)$ are given by $H \subseteq \hat{H}_\omega$ and a 2-cocycle of $H$. This is because when $\omega|_H = 1$, Eq.(27) is a 2-cocycle condition for the $L$-symbols: $L_{h_2,h_3}^x \cdot L_{h_1,h_2h_3}^x = L_{h_1,h_2}^x \cdot L_{h_1h_2,h_3}^x$. For a trivial 3-cocycle this results in the classification of SPT phases [7].

## 4.2 Symmetry action on domain walls and their interchange statistics

We consider now domain wall excitations between the ground states, described by $|\mathcal{I}|^2$ MPS tensors $\{e_{xy} | x, y \in \mathcal{I}\}$. A state with periodic boundary conditions hosting a pair of domain walls between the states $x$ and $y$ can be written as:

$$|\psi(x,y)\rangle = \sum_{\{i_k\}} \text{Tr}[e_{xy}^{i_1} A_y^{i_2} \cdots A_y^{i_{l-1}} e_{yx}^{i_l} A_x^{i_{l+1}} \cdots A_x^{i_N}] |i_1 \cdots i_{l-1} i_l i_{l+1} \cdots i_N\rangle.$$

The main assumption here is that the MPUs permute between the different domain walls, that is $U_g |\psi(x,y)\rangle \propto |\psi(gx,gy)\rangle$. Then, due to the injectivity of the MPS tensors and the relations in Eq. (26) we obtain that there are phase factors, denoted by $B_{x,y}^g$, which encode this permutation locally:

$$gy \quad \begin{matrix} g & g \\ & y & \\ & e_{yx} & \end{matrix} \quad gx = B_{y,x}^g \quad gy \quad \begin{matrix} & \\ e_{gygx} \end{matrix} \quad gx \quad \text{where} \quad B_{y,x}^g \sim \frac{\gamma_{g,x}}{\gamma_{g,y}} \cdot B_{y,x}^g. \tag{28}$$

These phase factors are subjected to certain relations with the $L$-symbols. Those can be obtained by comparing the two ways of locally decomposing the action of two group elements into an individual domain wall excitation, which results in

$$B_{hx,hy}^g B_{x,y}^h = \frac{L_{g,h}^x}{L_{g,h}^y} B_{x,y}^{gh}, \quad \forall g, h \in G, \quad \text{and} \quad x, y \in \mathcal{I}. \tag{29}$$

This equation can be seen as the fractionalization of the global symmetry on the domain walls. We remark that this equation is valid both for local excitations (where $x = y$ so they transform linearly) and for on-site symmetries.

Domain walls can be created on the physical Hilbert space with finite strings of MPUs terminated by an appropriate tensor. We assume that there is always a group element $g \in G$ permuting $x$ to $y$, such that $y = gx$. Then we can denote the operator creating a pair of domain walls, at sites $i$ and $j$ and from $x$ to $y = gx$, by the operator $O_g^{[i,j]}$ defined by:

$$|\psi(x,y)\rangle = O_g^{[i,j]}|\psi_{A_x}\rangle = \cdots \quad\quad \cdots ,\qquad (30)$$

where the yellow endpoint tensor is defined as follows:

$$\diamond = \sum_{x \in \mathcal{I}} \quad . \qquad (31)$$

We would like to study the properties of isolated domain walls. To this end, we cut the operator $O_g^{[i,j]}$ in the middle and define the following operator:

$$O_{g,a}^{[i]} = \quad |a\rangle \; . \qquad (32)$$

Then, comparing the effect of $O_g^{[i]}O_g^{[j]}$ versus $O_g^{[j]}O_g^{[i]}$ we get:

$$O_{g,a}^{[j]} \cdot O_{g,b}^{[i]}|\psi_x\rangle = B_{x,gx}^g \, O_{g,a}^{[i]} \cdot O_{g,b}^{[j]}|\psi_x\rangle, \quad i > j,$$

where we can interpret this identity as $B_{x,gx}^g$ being the phase factor from the interchange of two $g$-domain walls on top of the ground state $|\psi_x\rangle$. We note the swap on the virtual level of the operators and that $B_{x,gx}^g$ is gauge dependent.

## 4.3 Quantities to determine the quantum phase

We consider a quantum phase of $(G, \omega)$ characterized by $\{H \subseteq G : \omega_H = 1, \alpha \in \mathcal{H}^2[H, U(1)]\}$. We first show that the non-trivial part of $\omega$, given by the elements of $G \setminus H$, is captured by a gauge invariant quantity resulting from the appropriate interchange of several domain walls. Then we show that the 2-cocycle of $H$ also characterizes the action of the MPU on the domain walls.

These findings translate the characterization of quantum phases based on ground states (the $L$-symbols) into its characterization through the first excited states that could be detected by the dynamics of the system.

Let us consider a group element $g \in G \setminus H$ with order $o(g)$ ($gx \neq x$ for all $x \in \mathcal{I}$ and $g^{o(g)} = e$). The interchange of $o(g) + 1$ of these domain wall operators ordered by the sites $k_i > k_{i+1}$ and permuted from the first to the last, results in:

$$O_{g,c}^{[k_{o(g)+1}]} \cdots O_{g,b}^{[k_2]}O_{g,a}^{[k_1]}|\psi_x\rangle = B_{x,gx}^g O_{g,c}^{[k_{o(g)+1}]} \cdots O_{g,b}^{[k_1]}O_{g,a}^{[k_2]}|\psi_x\rangle$$

$$= \left(\prod_{i=1}^{o(g)} B_{g^i x, g^{i-1} x}^g\right) O_{g,c}^{[k_1]}O_{g,b}^{[k_{o(g)+1}]} \cdots O_{g,a}^{[k_2]}|\psi_x\rangle .$$

The phase factor $\prod_{i=1}^{o(g)} B_{g^i x, g^{i-1} x}^g$ is a gauge invariant quantity since a phase factor modification of the action tensors corresponds to $\prod_{i=1}^{o(g)} \frac{\gamma_{g,g^{i-1}x}}{\gamma_{g,g^i x}} = 1$. Using Eq. (29), the phase factor that appears in the interchange is

$$\prod_{i=1}^{o(g)} B_{g^i x, g^{i-1} x}^g = \prod_{i=1}^{o(g)-1} \frac{L_{g,g^i}^{gx}}{L_{g,g^i}^x} = \prod_{i=1}^{o(g)} \omega^{-1}(g, g^i, g), \qquad (33)$$

where in the second equality we have used Eq.(27) in the form

$$L_{g,g^i}^{gx}/L_{g,g^i}^{x} = \omega^{-1}(g,g^i,g)L_{g^i,g}^{x}/L_{g^{i+1},g}^{x},$$

using that $gx \neq x$ (since $\omega$ restricted to the unbroken symmetry group is trivial) Eq. (33) shows how the gauge invariant phase factor characterizing the interchange of domain walls can be written in terms of the 3-cocycle of the MPU symmetry and viceversa.

Notice that for trivial 3-cocycle, like on-site symmetries, the phase factor interchange is trivial. Moreover, this gauge invariant phase factor is independent of the ground state we start with.

Given $H$, the unbroken symmetry group $H = \{h \in G, | hx = x \text{ for all } x \in \mathcal{I}\}$, Eq.(29) restricted to $h_1, h_2 \in H$ has the following form:

$$B_{y,x}^{h_1} B_{y,x}^{h_2} = \frac{L_{h_1,h_2}^{y}}{L_{h_1,h_2}^{x}} B_{y,x}^{h_1 h_2},$$

where the factor $\frac{L_{h_1,h_2}^{y}}{L_{h_1,h_2}^{x}}$ satisfies a 2-cocycle condition, using Eq.(27) and Eq.(25), for every pair $y, x \in \mathcal{I}$:

$$\frac{L_{h_1,h_2}^{y}}{L_{h_1,h_2}^{x}} \frac{L_{h_1 h_2,h_3}^{y}}{L_{h_1 h_2,h_3}^{x}} = \frac{L_{h_1,h_2 h_3}^{y}}{L_{h_1,h_2 h_3}^{x}} \frac{L_{h_2,h_3}^{y}}{L_{h_2,h_3}^{x}},$$

with phase factor freedom $\frac{(\gamma_{h_1,y}/\gamma_{h_1,x}) \cdot (\gamma_{h_2,y}/\gamma_{h_2,x})}{(\gamma_{h_1 h_2,y}/\gamma_{h_1 h_2,x})}$, which shows that the action of the unbroken symmetry on the domain walls can be projective. Local excitations, $y = x$ so no domain wall, transform linearly: $B_{x,x}^{h_1} B_{x,x}^{h_2} = B_{x,x}^{h_1 h_2}$. This matches with the well known fact that for on-site symmetries local excitations can be decomposed into irreps of the symmetry group. Notice that Eq. (33) is also consistent with the following fact: if $gx = x$, such that $O_g |\psi_x\rangle$ is a local excitation and not a domain wall, then $L_{g,g^i}^{gx} = L_{g,g^i}^{x}$, that is, the phase factor there is trivial.

## 4.4 Example: $G = \mathbb{Z}_n$ with fully symmetry breaking

In this section we consider MPU symmetries coming from the abelian group $\mathbb{Z}_n$. The are $n$ distinct MPU representations according to its 3-cocycle since $\mathcal{H}^3(\mathbb{Z}_n, U(1)) = \mathbb{Z}_n$. The explicit formula is $\omega_j(a,b,c) = \exp\{\frac{2\pi i j}{n^2} a(b + c - [b + c])\}$, where $[b + c] = b + c \mod n$, and $j = 0, \cdots, n-1$ labels the different cocycle classes ($j = 0$ corresponds to the trivial 3-cocycle). Let us consider the maximally symmetry-broken phase, i.e. $n$-fold degeneracy, for every cocycle class $\omega_j$. Then, we can give a closed formula for the phase factor of the interchange of domain walls created from the group element $a$ in Eq.(33):

$$\prod_{k=1}^{o(a)} \omega_j^{-1}(a, a^k, a) = \exp\left\{-\frac{2\pi i j}{n^2} a \sum_{k=0}^{o(a)-1} (ka + a - [ka + a])\right\}.$$

Specifically, for $n = 3$, we obtain

| $a$ | 1 | 2 |
|---|---|---|
| $j = 1$ | $e^{-\frac{2\pi}{3}i}$ | $e^{-\frac{2\pi}{3}i}$ |
| $j = 2$ | $e^{\frac{2\pi}{3}i}$ | $e^{\frac{2\pi}{3}i}$ |

(34)

and for $n = 4$

| $a$ | 1 | 2 | 3 |
|---|---|---|---|
| $j = 1$ | $-i$ | $-1$ | $-i$ |
| $j = 2$ | $-1$ | $1$ | $-1$ |
| $j = 3$ | $i$ | $-1$ | $i$ |

(35)

From these tables, we can infer that the excitations exhibit a statistics which is neither fermionic nor bosonic.

## 5 Detection of the phase factors

In this section we propose a finite size operator whose expectation value on a ground state, labeled by $x$, returns in the gauge invariant quantity of Eq.(33); it can thus be used to measure the domain wall statistics and thereby detect the nature of the underlying phase. The construction of the operator is based on truncating the MPU symmetry. It does not require an MPS representation of the ground state nor the explicit form of the domain wall excitations.

We will start by discussing the $\mathbb{Z}_2$ case where the goal is to reproduce Eq. (23) and get $c_{AB}c_{BA} = \omega$, see Eq.(17). The first step is to consider a truncated version of the MPU symmetry $U$, restricted to a region which includes an interval $[i_1, j_1]$ and a region on the scale of the correlation length around it; we will denote this truncated unitary by $U^{[i_1,j_1]}$. We can always construct such a truncated $U^{[i_1,j_1]}$, since any MPU can be rewritten as a unitary circuit of depth 2 with nearest-neighbor gates under finite blocking [11]; we can then obtain $U^{[i_1,j_1]}$ by dropping the gates outside the given range. The resulting $U^{[i_1,j_1]}$ is then a unitary operator which is equal to $U$ inside the given region. Thus, the action of $U^{[i_1,j_1]}$ on a ground state $A$ produces the same effect inside the region as $U$ – that is, it exchanges $A$ and $B$ if the distance between $i_1$ and $j_1$ is bigger than the correlation length, and acts trivially outside the region. Therefore, $U^{[i_1,j_1]}$ creates a pair of domain walls around sites $i_1$ and $j_1$.

If we act with $U$ on top of $U^{[i_1,j_1]}$, we will obtain the global phase factor $c_{BA}c_{AB}$. Since we want to obtain this phase factor by using operators with a finite range, we instead use $U^{[i_2,j_2]}$, which creates a pair of domain walls around sites $i_2$ and $j_2$, where we choose $i_2 < i_1 < j_1 < j_2$, and demand that the distance between the different positions is sufficiently bigger than the length scale set by the correlation length of the ground state.

To get a non-vanishing expectation value on a ground state, we need to 'undo' the domain walls and map them back to the ground space. To achieve this, we use first $(U^{[i_1,j_1]})^\dagger$ and then $(U^{[i_2,j_2]})^\dagger$. Notice that in general $(U_g^{[i_1,j_1]})^\dagger$ and $(U_g^\dagger)^{[i_1,j_1]}$ are not the same, but are related by unitaries at the boundaries. The result can be written as:

$$\langle\psi_A|(U^{[i_2,j_2]})^\dagger(U^{[i_1,j_1]})^\dagger U^{[i_2,j_2]}U^{[i_1,j_1]}|\psi_A\rangle = \omega, \tag{36}$$

which precisely measures the commutator between $U^{[i_1,j_1]}$ and $U^{[i_2,j_2]}$.

As an aside, note that for on-site symmetries on a chain $\Lambda$, truncating the global symmetry $U = \bigotimes_{i\in\Lambda} u^{[i]}$ results in $U^{[a,b]} = \bigotimes_{i\in[a,b]} u^{[i]}$. Therefore, for on-site symmetries we get that $(U^{[i_2,j_2]})^\dagger(U^{[i_1,j_1]})^\dagger U^{[i_2,j_2]}U^{[i_1,j_1]} = \mathbb{1}$, so its expectation value is $+1$, in agreement with our findings in the previous sections.

For the general case of any finite group $G$, the first thing to notice is that the phase factor $\prod_{i=1}^{o(g)} B_{g^i x, g^{i-1} x}^g$ of Eq.(33) only depends on the order of the element $g$ (and its powers) and not on the group itself. That reduces the problem of truncating MPUs representing the cyclic group $\mathbb{Z}_{o(g)}$. We first create a pair of domain walls around $i_0$ and $j_0$ by acting with the truncated unitary $U_g^{[i_0,j_0]}$ on the ground state labeled by $x$. We will focus on the excitation created around $j_0$ which correspond to a domain wall of type $gx - x$ (analogous to the tensor $e_{gx,x}$ in the MPS scenario). We now apply $U_g^{[i_1,j_1]}$ where $i_0 < i_1 < j_0 < j_1$, obtaining the phase factor $B_{gx,x}^g$ and permuting the domain wall type to $g^2 x - gx$, see (28). We have also created domain walls around $i_1$ and $j_1$ but we will avoid acting on them with the next truncated unitaries. To achieve this, we act with $U_g^{[i_2,j_2]}$ satisfying $i_1 < i_2 < j_0 < j_2 < j_1$, where we get $B_{g^2 x, gx}^g$ and the domain wall is now of type $g^3 x - g^2 x$. Following this strategy we can conclude that the phase

factor of Eq.(33) can be obtained by applying the following operator:

$$X = \prod_{i=0}^{o(g)} U_g^{[i_i, j_i]}, \quad i_0 < i_{o(g)} < \cdots < i_1 < j_0 < j_{o(g)} < \cdots < j_1.$$

Finally, to get a non-vanishing expectation value on a ground state, we need to annihilate the domain wall excitations. The important point here is to 'undo' each of the domain walls without acting on any of the other ones, which would result in undesirable $B$ phases. This can be achieved by acting with

$$Y = (U^{[i_0, j_1]})^\dagger (U^{[i_{o(g)}, j_2]})^\dagger \cdots (U^{[i_2, j_{o(g)}]})^\dagger (U^{[i_1, j_0]})^\dagger.$$

Finally:

$$\langle \psi_x | YX | \psi_x \rangle = \prod_{i=1}^{o(g)} B_{g^i x, g^{i-1} x}^g. \tag{37}$$

We notice that the previous method is also valid for $\mathbb{Z}_2$, we would obtain $B_{gx,x}^g B_{x,gx}^g \equiv c_{AB} c_{BA} = \omega$, but it does not coincide with the one presented in Eq.(36). This is because for Eq.(36) we have used that $g^{-1} = g$ so $B_{x,g^{-1}x}^g = B_{x,gx}^g \equiv c_{AB}$, so this phase factor can be obtained by acting with $g$ on the domain wall $x - gx$ (the partner of the domain wall $gx - x$) and then simplify the general procedure.

# 6 Conclusions

We have studied the exchange statistics of domain wall excitations in gapped spin chains with anomalous symmetry breaking, and have found that the anomaly of the symmetry and the way in which the symmetry broken ground states transform under it is directly reflected in the statistics of the domain wall excitations. Our results extend Ref. [14], which characterized the anomalous gapped phases through their ground states, beyond ground state properties, by connecting them to the properties exhibited by their low-lying excited states and thus to to physically testable properties.

Our main technical tools are the representation of the ground states by injective MPS, and the characterization of the anomalous symmetry action on them through local fusion tensors. This allowed us to construct domain wall excitations in terms of the ground state MPS and the operators creating them in terms of MPUs, which in turn enabled us to extract their exchange statistics from the fusion tensors.

The approach pursued here can be extended to non-invertible global symmetries representing fusion categories in terms of matrix product operators, whose gapped phases have been classified in Ref. [14] as well. Such a tensor network approach will provide a local characterization of the boundary excitation of string-net models developed by Kitaev and Kong in Ref. [30]. It could also be interesting to generalized our study to anomalous non-internal symmetries like time reversal or reflexion.

# Acknowledgments

JGR thanks Laurens Lootens for helpful conversations.

**Funding information** This research was funded in part by the European Union's Horizon 2020 research and innovation programme through Grant No. 863476 (ERC-CoG SEQUAM), the Austrian Science Fund FWF (Grant DOIs doi:10.55776/COE1, doi:10.55776/P36305, and doi:10.55776/F71), and the European Union through NextGenerationEU. For open access purposes, the authors have applied a CC BY public copyright license to any author accepted manuscript version arising from this submission.

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
