# Peer review of "Fractional domain wall statistics in spin chains with anomalous symmetries"

_SciPost Physics, doi:SciPost Phys. 18, 043 (2025)_

## Round 1 · Referee Report · Anonymous (Referee 1) · 2024-11-20

Strengths

  • The interplay between symmetry anomalies and topological features in low-dimensional quantum systems represent a novel and timely topic in condensed matter physics

  • The use of tensor networks (MPS and MPUs) is rigorous and builds on a well-established framework

  • The paper both general and specific cases, providing clarity on how anomalies manifest in exchange statistics.

  • The paper is well-organized, with detailed mathematical derivations.

Weaknesses

  • The paper is largely theoretical and could potentially benefit from accompanying numerical simulations (e.g. by tuning away from fixed points of the model and measuring the finite sie operator proposed in Sec. V).

Report

The paper investigates the exchange statistics of domain wall excitations in one-dimensional (1D) quantum spin chains with anomalous symmetries, utilizing Matrix Product States (MPS) and Matrix Product Unitaries (MPU) as key theoretical frameworks. It provides a connection between the anomaly of the symmetry and the physical properties of domain wall excitations, presenting a method to measure these properties experimentally. The work extends the classification of anomalous gapped phases beyond ground state properties and links them to physically measurable quantities in low-energy excitations.

Requested changes

  • It might be helpful to clarify the emergence of anyons in 1D and their stability. Generally, one would expect that anyons can only emerge in presence of topological order, which is absent in 2D.

  • Maybe there are some typos following Eq. (1): Should it be Z_i = Z_j = 1 instead of I=j=1? Also, should it read \mu > 1 instead of \mu>0?

Recommendation

Publish (easily meets expectations and criteria for this Journal; among top 50%)

---

## Round 1 · Referee Report · Anonymous (Referee 2) · 2024-12-16

Strengths

  • Generality of the method

  • Clarity of the presentation

Report

This paper clarifies the relationship between the statistics of domain-wall excitations and the anomaly of symmetries in one-dimensional systems.

Requested changes

1. Throughout the manuscript, the notation for the system size alternates between n and N. These should be unified.

2. Regarding Eq. (6), is $m >\geq0$ a typo?

3. For the domain wall tensor $e_{AB}^i$, is there a necessity for the leg $i$ to have the same dimension as the physical leg?

4. Regarding Eq. (8), the MPS tensors from site $1$ to $k$ are missing.

5. Regarding Eq. (10), while this equation seems to be a sufficient condition, is it also a necessary condition? Could you provide the derivation of Eq.(10)?

6. In Phys. Rev. B 107, 155136, they introduce symmetry operators truncated to a segment (referred to as “patch operators”) and specifically calculate the F-symbol for anomalous $Z_2$ symmetry. Since the calculation for $Z_2$ symmetry appears to be essentially the same, wouldn’t it be appropriate to cite this work?

Recommendation

Publish (easily meets expectations and criteria for this Journal; among top 50%)

---

## Editorial Decision

published